# Convolution with even-sized kernels and symmetric padding

**Shuang Wu**[1], **Guanrui Wang**[1], **Pei Tang**[1], **Feng Chen**[2], **Luping Shi**[1]
[1]Department of Precision Instrument, [2]Department of Automation
Center for Brain Inspired Computing Research
Beijing Innovation Center for Future Chip
Tsinghua University
{lpshi,chenfeng}@mail.tsinghua.edu.cn

## Abstract

Compact convolutional neural networks gain efficiency mainly through depthwise convolutions, expanded channels and complex topologies, which contrarily aggravate the training process. Besides, $3\times3$ kernels dominate the spatial representation in these models, whereas even-sized kernels ($2\times2$, $4\times4$) are rarely adopted. In this work, we quantify the shift problem occurs in even-sized kernel convolutions by an information erosion hypothesis, and eliminate it by proposing symmetric padding on four sides of the feature maps (C2sp, C4sp). Symmetric padding releases the generalization capabilities of even-sized kernels at little computational cost, making them outperform $3\times3$ kernels in image classification and generation tasks. Moreover, C2sp obtains comparable accuracy to emerging compact models with much less memory and time consumption during training. Symmetric padding coupled with even-sized convolutions can be neatly implemented into existing frameworks, providing effective elements for architecture designs, especially on online and continual learning occasions where training efforts are emphasized.

## 1   Introduction

Deep convolutional neural networks (CNNs) have achieved significant successes in numerous computer vision tasks such as image classification [37], semantic segmentation [43], image generation [8], and game playing [28]. Other than domain-specific applications, various architectures have been designed to improve the performance of CNNs [11, 14, 3], wherein the feature extraction and representation capabilities are mostly enhanced by deeper and wider models containing ever-growing numbers of parameters and operations. Thus, the memory overhead and computational complexity greatly impede their deployment in embedded AI systems. This motivates the deep learning community to design compact CNNs with reduced resources, while still retaining satisfactory performance.

Compact CNNs mostly derive generalization capabilities from architecture engineering. Shortcut connection [11] and dense concatenation [14] alleviate the degradation problem as the network deepens. Feature maps (FMs) are expanded by pointwise convolution (C1) and bottleneck architecture [35, 40]. Multi-branch topology [38], group convolution [42], and channel shuffle operation [47] recover accuracy at the cost of network fragmentation [25]. More recently, there is a trend towards mobile models with <10M parameters and <1G FLOPs [13, 25, 23], wherein the depthwise convolution (DWConv) [5] plays a crucial role as it decouples cross-channel correlations and spatial correlations. Aside from human priors and handcrafted designs, emerging neural architecture search (NAS) methods optimize structures by reinforcement learning [48], evolution algorithm [32], etc.

Despite the progress, the fundamental spatial representation is dominated by $3\times3$ kernel convolutions (C3) and the exploration of other kernel sizes is stagnating. Even-sized kernels are deemed inferior and

rarely adopted as basic building blocks for deep CNN models [37, 38]. Besides, most of the compact models concentrate on the inference efforts (parameters and FLOPs), whereas the training efforts (memory and speed) are neglected or even becoming more intractable due to complex topologies [23], expanded channels [35], and additional transformations [15, 47, 40]. With the growing demands for online and continual learning applications, the training efforts should be jointly addressed and further emphasized. Furthermore, recent advances in data augmentation [46, 7] have shown much more powerful and universal benefits. A simpler structure combined with enhanced augmentations easily eclipses the progress made by intricate architecture engineering, inspiring us to rethink basic convolution kernels and the mathematical principles behind them.

In this work, we explore the generalization capabilities of even-sized kernels ($2\times2$, $4\times4$). Direct implementation of these kernels encounters performance degradation in both classification and generation tasks, especially in deep networks. We quantify this phenomenon by an *information erosion hypothesis*: even-sized kernels have asymmetric receptive fields (RFs) that produce pixel shifts in the resulting FMs. The location offset accumulates when stacking multiple convolutions, thus severely eroding the spatial information. To address the issue, we propose convolution with even-sized kernels and symmetric padding on each side of the feature maps (C2sp, C4sp).

Symmetric padding not merely eliminates the shift problem, but also extends RFs of even-sized kernels. Various classification results demonstrate that C2sp is an effective decomposition of C3 in terms of 30%-50% saving of parameters and FLOPs. Moreover, compared with compact CNN blocks such as DWConv, inverted-bottleneck [35], and ShiftNet [40], C2sp achieves competitive accuracy with >20% speedup and >35% memory saving during training. In generative adversarial networks (GANs) [8], C2sp and C4sp both obtain stabilized convergence and improved image qualities. Our work stimulates a new perspective full of optional units for architecture engineering, as well as provides basic but effective alternatives that balance both the training and inference efforts.

## 2   Related work

Our method belongs to compact CNNs that design new architectures and then train them from scratch. Whereas most network compressing methods in the literature attempt to prune weights [9] or quantize operands and operations [41] in terms of implementation on neural accelerators [4, 31]. These methods are orthogonal to our work and can be jointly adopted.

**Even-sized kernel** Even-sized kernels are mostly applied together with stride 2 to resize images. For example, GAN models in [27] apply $4\times4$ kernels and stride 2 in the discriminators and generators to avoid the checkerboard artifact [29]. However, $3\times3$ kernels are preferred when it comes to deep and large-scale GANs [18, 21, 3]. Except for scaling, few works have implemented even-sized kernels as basic building blocks for their CNN models. In [10], $2\times2$ kernels are tested with relatively shallow (about 10 layers) models, where the FM sizes between two convolution layers are not preserved strictly. In relational reinforcement learning [45], two C2 layers are adopted to achieve reasoning and planning of objects represented by 4 pixels.

**Atrous convolution** Dilated convolution [43] supports exponential expansions of RFs without loss of resolution or coverage, which is specifically suitable for dense prediction tasks such as semantic segmentation. Deformable convolution [6] augments the spatial sampling locations of kernels by additional 2D offsets and learning the offsets directly from target datasets. Therefore, deformable kernels shift at pixel-level and focus on geometric transformations. ShiftNet [40] sidesteps spatial convolutions entirely by shift kernels that contain no parameter or FLOP. However, it requires large channel expansions to reach satisfactory performance. Active shift [16] formulates the amount of FM shift as a learnable function and optimized two parameters through end-to-end backpropagation.

## 3   Symmetric padding

### 3.1   The shift problem

We start with the spatial correlation in basic convolution kernels. Intuitively, replacing a C3 with two C2s should provide performance gains aside from an 11% reduction of overheads, which is inspired by the factorization of C5 into two C3s [37]. However, experiments in Figure 3 indicate that the classification accuracy of C2 is inferior to C3 and saturate much faster as the network deepens.

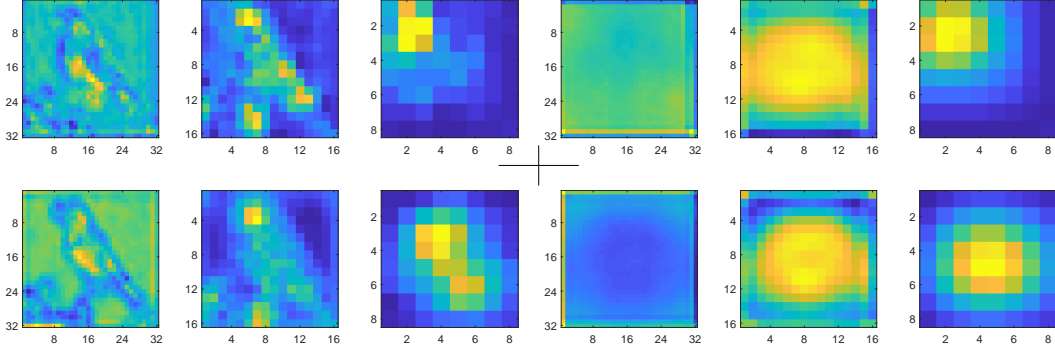

Figure 1: Normalized FMs derived from well-trained ResNet-56 models. Three spatial sizes $32{\times}32$, $16{\times}16$, and $8{\times}8$ before down-sampling stages are presented. First row: Conv2$\times$2 with asymmetric padding (C2). Second row: Conv2$\times$2 with symmetric padding (C2sp). Left: a sample in the CIFAR10 test dataset. Right: average results from all the samples in the test dataset.

Besides, replacing each C3 with C4 also hurts accuracy even though a $3{\times}3$ kernel can be regarded as a subset of $4{\times}4$ kernel, which contains 77% more parameters and FLOPs. To address this issue, the FMs of well-trained ResNet-56 [11] models with C2 and C2sp are visualized in Figure 1. Since every single FM (channel) is very stochastic and hard to interpret, we report the average values of all channels in that convolution layer. FMs of C4 and C3 have similar manners as C2 and C2sp, respectively and are omitted for clarity. It is clearly seen that the post-activation (ReLU) values in C2 are gradually shifting to the left-top corner of the spatial location. These compressed and distorted features are not suitable for the following classification, let alone pixel-level tasks based on it such as detection and semantic segmentation, where all the annotations will have offsets starting from the left-top corner of the image.

We identify this as the *shift problem* observed in even-sized kernels. For a conventional convolution between $c_i$ input and $c_o$ output FMs $\mathcal{F}$ and square kernels $\boldsymbol{w}$ of size $k \times k$, it can be given as

$$\mathcal{F}^o(\boldsymbol{p}) = \sum_{i=1}^{c_i} \sum_{\boldsymbol{\delta} \in \mathcal{R}} \boldsymbol{w}_i(\boldsymbol{\delta}) \cdot \mathcal{F}^i(\boldsymbol{p} + \boldsymbol{\delta}), \tag{1}$$

where $\boldsymbol{\delta}$ and $\boldsymbol{p}$ enumerate locations in RF $\mathcal{R}$ and in FMs of size $h \times w$, respectively. When $k$ is an odd number, e.g., 3, we define the central point of $\mathcal{R}$ as origin:

$$\mathcal{R} = \{(-\kappa, -\kappa), (-\kappa, 1-\kappa), \dots, (\kappa, \kappa)\}, \kappa = \lceil \frac{k-1}{2} \rceil, \tag{2}$$

where $\kappa$ denotes the maximum pixel number from four sides to the origin. $\lceil \cdot \rceil$ is the ceil rounding function. Since $\mathcal{R}$ is symmetrical, we have $\sum_{\boldsymbol{\delta} \in \mathcal{R}} \boldsymbol{\delta} = (0, 0)$.

When $k$ is an even number, e.g., 2 or 4, implementing convolution between $\mathcal{F}^i$ and kernels $\boldsymbol{w_i}$ becomes inevitably asymmetric since there is no central point to align. In most deep learning frameworks, it draws little attention and is obscured by pre-defined offsets. For example, TensorFlow [1] picks the nearest pixel in the left-top direction as the origin, which gives an asymmetric $\mathcal{R}$:

$$\mathcal{R} = \{(1-\kappa, 1-\kappa), (1-\kappa, 2-\kappa), \dots, (\kappa, \kappa)\}, \sum_{\boldsymbol{\delta} \in \mathcal{R}} \boldsymbol{\delta} = (\kappa, \kappa). \tag{3}$$

The shift occurs at all the spatial locations $\boldsymbol{p}$ and is equivalent to pad one more zero on the bottom and right sides of FMs before convolutions. On the contrary, Caffe [17] pads one more zero on the left and top sides. PyTorch [30] only supports symmetric padding by default, users need to manually define the padding policy if desired.

## 3.2 The information erosion hypothesis

According to the above, even-sized kernels make zero-padding asymmetric with 1 pixel, and averagely (between two opposite directions) lead to 0.5-pixel shifts in the resulting FMs. The position offset

accumulates when stacking multiple layers of even-sized convolutions, and eventually squeezes and distorts features to a certain corner of the spatial location. Ideally, in case that such asymmetric padding is performed for $n$ times in the TensorFlow style with convolutions in between, the resulting pixel-to-pixel correspondence of FMs will be

$$\mathcal{F}_n\left[\boldsymbol{p} - (\frac{n}{2}, \frac{n}{2})\right] \leftarrow \mathcal{F}_0(\boldsymbol{p}). \tag{4}$$

Since FMs have finite size $h \times w$ and are usually down-sampled to force high-level feature representations, then the *edge effect* [26, 2] cannot be ignored because zero-padding at edges will distort the effective values of FM, especially in deep networks and small FMs. We hypothesize that the quantity of information $\mathcal{Q}$ is equal to the mean L1-norm of the FM, then successive convolutions with zero-padding to preserve FM size will gradually erode the information:

$$\mathcal{Q}_n = \frac{1}{hw} \sum_{\boldsymbol{p} \in h \times w} |\mathcal{F}_n(\boldsymbol{p})|, \quad \mathcal{Q}_n < \mathcal{Q}_{n-1}. \tag{5}$$

The *information erosion* happens recursively and is very complex to be formulated, we directly derive FMs from deep networks that contain various kernel sizes. In Figure 2, 10k images of size $32 \times 32$ are fed into untrained ResNet-56 models where identity connections and batch normalizations are removed. $\mathcal{Q}$ decreases progressively and faster in larger kernel sizes and smaller FMs. Besides, asymmetric padding in even-sized kernels (C2, C4) speeds up the erosion dramatically, which is consistent with well-trained networks in Figure 1. An analogy is that FM can be seen as a rectangular ice chip melting in water except that it can only exchange heat on its four edges. The smaller the ice, the faster the melting process happens. Symmetric padding equally distributes thermal gradients so as to slow down the exchange. Whereas asymmetric padding produces larger thermal gradients on a certain corner, thus accelerating it.

The hypothesis also provides explanations for some experimental observations in the literature. (1) The degradation problem in very deep networks [11]: although the vanishing/exploding forward activations and backward gradients have been addressed by intermediate normalization [15], the spatial information is eroded and blurred by the edge effect after multiple convolutions. (2) It is reported [3] that in GANs, doubling the depth of networks hampers training, and increasing the kernel size to 5 or 7 leads to minor improvement or even degradation. These indicate that GANs require augmented spatial information and are more sensitive to progressive erosion.

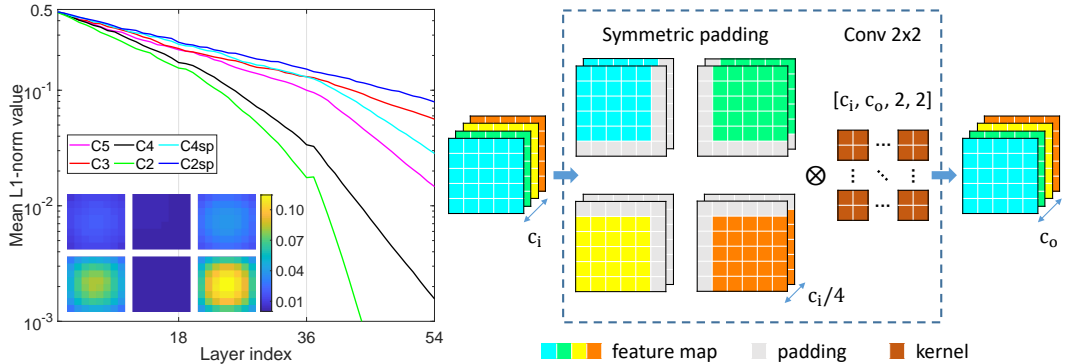

Figure 2: Left: layerwise quantity of information $\mathcal{Q}$ and colormaps derived from the last convolution layers. FMs are down-sampled after 18th and 36th layers. Right: implementation of convolution with $2 \times 2$ kernels and symmetric padding (C2sp).

## 3.3  Method

Since $\mathcal{R}$ is inevitably asymmetric for even kernels in Equation 3, it is difficult to introduce symmetry within a single FM. Instead, we aim at the final output $\mathcal{F}^o$ summed by multiple input $\mathcal{F}^i$ and kernels. For clarity, let $\mathcal{R}_0$ be the shifted RF in Equation 3 that picks the nearest pixel in the left-top direction as origin, then we explicitly introduce a shifted collection $\mathcal{R}_+$

$$\mathcal{R}_+ = \{\mathcal{R}_0, \mathcal{R}_1, \mathcal{R}_2, \mathcal{R}_3\} \tag{6}$$

that includes all four directions: left-top, right-top, left-bottom, right-bottom.

Let $\pi : I \to \mathcal{R}_+$ be the surjective-only mapping from input channel indexes $i \in I = \{1, 2, ..., c_i\}$ to certain shifted RFs. By adjusting the proportion of four shifted RFs, we can ensure that

$$\sum_{i=1}^{c_i} \sum_{\boldsymbol{\delta} \in \pi(i)} \boldsymbol{\delta} = (0, 0). \tag{7}$$

When mixing four shifted RFs within a single convolution, the RFs of even-sized kernels are partially extended, e.g., $2\times2 \to 3\times3$, $4\times4 \to 5\times5$. If $c_i$ is an integer multiple of 4 (usually satisfied), the symmetry is strictly obeyed within a single convolution layer by distributing RFs in sequence

$$\pi(i) = \mathcal{R}_{\lfloor 4i/c_i \rfloor}. \tag{8}$$

As mention above, the shifted RF is equivalent to pad one more zero at a certain corner of FMs. Thus, the symmetry can be neatly realized by a grouped padding strategy, an example of C2sp is illustrated in Figure 2. In summary, the 2D convolution with even-sized kernels and symmetric padding consists of three steps: (1) Dividing the input FMs equally into four groups. (2) Padding FMs according to the direction defined in that group. (3) Calculating the convolution without any padding. We have also done ablation studies on other methods dealing with the shift problem, please see Section 5.

## 4 Experiments

In this section, the efficacy of symmetric padding is validated in CIFAR10/100 [20] and ImageNet [33] classification tasks, as well as CIFAR10, LSUN bedroom [44], and CelebA-HQ [18] generation tasks. First of all, we intuitively demonstrate that the shift problem has been eliminated by symmetric padding. In the symmetric case of Figure 1, FMs return to the central position, exhibiting healthy magnitudes and reasonable geometries. In Figure 2, C2sp and C4sp have much lower attenuation rates than C2 and C4 regarding information quantity $\mathcal{Q}$. Besides, C2sp has larger $\mathcal{Q}$ than C3, expecting performance improvement in the following evaluations.

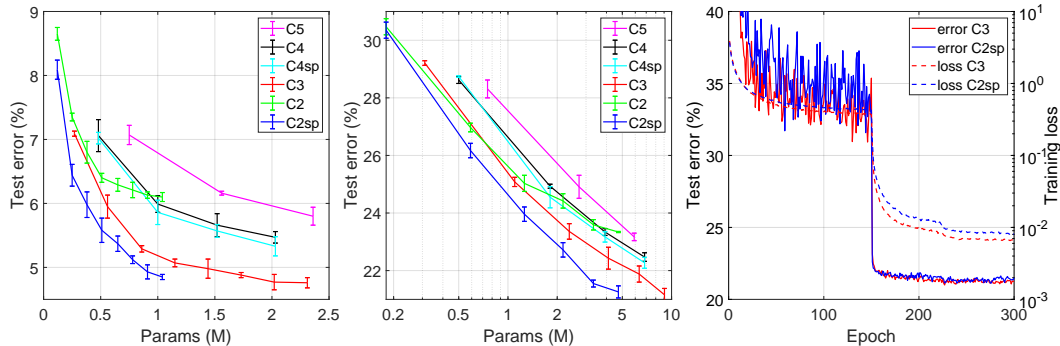

Figure 3: Left and middle: parameter-accuracy curves of ResNets and DenseNets that contain multiple depths and various convolution kernels. Right: training and testing curves on DenseNet-112 with C3 and C2sp.

### 4.1 Exploration of various kernel sizes

To explore the generalization capabilities of various convolution kernels, ResNet series without bottleneck architectures [11] are chosen as the backbones. We maintain all the other components and training hyperparameters as the same, and only replace each C3 by a C4, C2 or C2sp. The networks are trained on CIFAR10 with depths in $6n + 2, n \in \{3, 6, \ldots, 24\}$. The parameter-accuracy curves are shown in Figure 3. The original even-sized kernels $4\times4$, $2\times2$ perform poorly and encounter faster saturation as the network deepens. Compared with C3, C2sp reaches similar accuracy with only 60%-70% of the parameters, as well as FLOPs that are linearly correlated. We also find that symmetric padding only slightly improves the accuracy of C4sp. Considering the worse performance of C5 and the attenuation curves in Figure 2, the edge effect might dominate the information erosion of $4\times4$ kernels rather than the shift problem in such network depth.

Since the network architectures or datasets may affect the conclusion, we further evaluate these kernels on CIFAR100 with DenseNet [14] backbone to cross-validate the generality and consistency. The results of depths $6n + 4, n \in \{3, 6, \ldots, 18\}$ are shown in Figure 3. At the same depth, C2sp achieves comparable accuracy to C3 as the network gets deeper. The training losses indicate that C2sp have better generalization and less overfitting than C3. Under the criterion of similar accuracy, a C2sp model will save 30%-50% parameters and FLOPs in the CIFAR evaluations. Therefore, we recommend using C2sp as a better alternative to C3 in classification tasks.

## 4.2 Compare with compact CNN blocks

To facilitate fair comparisons for C2sp with compact CNN blocks that contain C1, DWConvs, or shift kernels, we use ResNets as backbones and adjust the width and depth to maintain the same number of parameters and FLOPs (overheads). In case there are $n$ input channels for a basic residual block, then two C2sp layers will consume about $8n^2$ overheads, the expansion is marked as 1-1-1 since no channel expands. For ShiftNet blocks [40], we choose expansion rate 3 and 3×3 shift kernels as suggested, the overheads are about $6n^2$. Therefore, the value of $n$ is slightly increased. While for the inverted-bottleneck [35], the suggested expansion rate 6 results in $12n^2 + O(6n)$ overheads, thus the number of blocks is reduced by $1/3$. For depthwise-separable convolutions [5], the overheads are about $2n^2 + O(n)$, so the channels are doubled and formed as 2-2-2 expansions.

Table 1: Comparison of various compact CNN blocks on CIFAR100. Shift, Invert, and Sep denotes ShiftNet block, inverted-bottleneck, and depthwise-separable convolution, respectively. *mixup* denotes training with mixup augmentation. Exp denotes the expansion rates of channels in that block. SPS refers to the speed during training: samples per second.

| Model | Block | Error (%) standard | Error (%) with *mixup* | Params (M) | FLOPs (M) | Exp | Memory (MB) | Speed (SPS) |
|---|---|---|---|---|---|---|---|---|
| 20 | Shift | $26.87 \pm 0.19$ | $24.74 \pm 0.04$ | 0.49 | 70.0 | 1-3-1 | 853 | 1906 |
| | Invert | $26.86 \pm 0.11$ | $25.32 \pm 0.12$ | 0.42 | 76.7 | 1-6-1 | 1219 | 2057 |
| | Sep | $26.70 \pm 0.20$ | $\mathbf{23.81 \pm 0.15}$ | 0.51 | 73.6 | 2-2-2 | 732 | 2709 |
| | C2sp | $26.77 \pm 0.19$ | $24.90 \pm 0.17$ | 0.50 | 73.2 | 1-1-1 | **487** | **3328** |
| 56 | Shift | $24.07 \pm 0.24$ | $21.31 \pm 0.21$ | 1.48 | 213.4 | 1-3-1 | 2195 | 683 |
| | Invert | $\mathbf{22.36 \pm 0.25}$ | $21.48 \pm 0.25$ | 1.52 | 240.1 | 1-6-1 | 2561 | 803 |
| | Sep | $23.31 \pm 0.09$ | $\mathbf{20.69 \pm 0.11}$ | 1.47 | 218.0 | 2-2-2 | 1707 | 1034 |
| | C2sp | $23.19 \pm 0.29$ | $20.91 \pm 0.22$ | 1.54 | 224.2 | 1-1-1 | **1219** | **1230** |
| 110 | Shift | $22.94 \pm 0.26$ | $20.47 \pm 0.18$ | 2.97 | 428.3 | 1-3-1 | 4146 | 348 |
| | Invert | $\mathbf{21.76 \pm 0.17}$ | $20.43 \pm 0.20$ | 3.17 | 485.1 | 1-6-1 | 4756 | 426 |
| | Sep | $22.31 \pm 0.22$ | $\mathbf{19.42 \pm 0.05}$ | 2.91 | 434.4 | 2-2-2 | 3170 | 540 |
| | C2sp | $21.93 \pm 0.04$ | $19.52 \pm 0.10$ | 3.10 | 450.7 | 1-1-1 | **1951** | **636** |

The results are summarized in Table 1. Since most models easily overfit CIFAR100 training set with standard augmentation, we also train the models with mixup [46] augmentation to make the differences more significant. In addition to error rates, the memory consumption and speed during training are reported. C2sp performs better accuracy than ShiftNets, which indicates that sidestepping spatial convolutions entirely by shift operations may not be an efficient solution. Compared with blocks that contain DWConv, C2sp achieves competitive results in 56 and 110 nets with fewer channels and simpler architectures, which reduce memory consumption (>35%) and speed up (>20%) the training process.

In Table 2, we compare C2sp with NAS models: NASNet [48], PNASNet [23], and AmoebaNet [32]. We apply Wide-DenseNet [14] and adjust the width and depth (K = 48, L = 50) to have approximately 3.3M parameters. C2sp suffers less than 0.2% accuracy loss compared with state-of-the-art auto-generated models, and achieves better accuracy (+0.21%) when the augmentation is enhanced. Although NAS models leverage fragmented operators [25], e.g., pooling, group convolution, DWConv to improve accuracy with similar numbers of parameters, the regular-structured Wide-DenseNet has better memory and computational efficiency in runtime. In our reproduction, the training speeds on TitanXP for NASNet-A and Wide-DesNet are about 200 and 400 SPS, respectively.

Table 2: Test error rates (%) on CIFAR10 dataset. c/o and *mixup* denotes cutout [7] and mixup [46] data augmentation.

| Model | Error (%) | Params (M) |
|---|---|---|
| NASNet-A [48] | 3.41 | 3.3 |
| PNASNet-5 [23] | 3.41 | 3.2 |
| AmoebaNet-A [32] | **3.34** | 3.2 |
| Wide-DenseNet C3 | 3.81 | 3.4 |
| Wide-DenseNet C2sp | 3.54 | 3.2 |
| NASNet-A + c/o [48] | 2.65 | 3.3 |
| Wide-DenseNet C2sp + c/o + *mixup* | **2.44** | 3.2 |

## 4.3 ImageNet classification

We start with the widely-used ResNet-50 and DenseNet-121 architectures. Since both of them contain bottlenecks and C1s to scale down the number of channels, C3 only consumes about 53% and 32% of the total overheads. Changing C3s to C2sp results in about 25% and 17% reduction of parameters and FLOPs, respectively. The top-1 classification error rates are shown in Table 3, C2sp have minor loss (0.2%) in ResNet, and slightly larger degradation (0.5%) in DenseNet. After all, there are only 0.9M parameters for spatial convolution in DenseNet-121 C2sp.

We further scale the channels of ResNet-50 down to 0.5× as a mobile setting. At this stage, a C2 model (asymmetric), as well as reproductions of MobileNet-v2 [35] and ShuffleNet-v2 [25] are evaluated. Symmetric padding greatly reduces the error rate of ResNet-50 0.5× C2 for 2.5%. On this basis, we propose an optimized ResNet-50 0.5× C2sp model that achieves comparable accuracy to compact CNNs, but has fewer parameters and FLOPs. Although MobileNet-v2 presents the best accuracy, it uses inverted-bottlenecks (the same structure in Table 1) to expand too many FMs, which significantly increase the memory consumption and slow down the training process (about 400 SPS), while other models can easily reach 900 SPS.

Table 3: Top-1 error rates on ImageNet. Results are obtained by our rseproductions using the same training hyperparameters.

| Model | Error (%) | Params (M) | FLOPs (M) |
|---|---|---|---|
| ResNet-50 C3 [11] | **23.8** | 25.5 | 4089 |
| ResNet-50 C2sp | 24.0 | 19.3 | 3062 |
| DenseNet-121 C3 [14] | 24.6 | 8.0 | 2834 |
| DenseNet-121 C2sp | 25.1 | **6.7** | **2143** |
| ResNet-50 0.5× C3 | **26.8** | 6.9 | 1127 |
| ResNet-50 0.5× C2 | 29.8 | 5.3 | 870 |
| ResNet-50 0.5× C2sp | 27.3 | **5.3** | **870** |
| ResNet-50 0.5× C2sp optim | 26.8 | **5.8** | **573** |
| ShuffleNet v2 2.0× [25] | 26.6 | 7.4 | 591 |
| MobileNet v2 1.4× [35] | **25.8** | 6.1 | 582 |

## 4.4 Image generation

The efficacy of symmetric padding is further validated in image generation tasks with GANs. In CIFAR10 32×32 image generation, we follow the same architecture described in [27], which has about 6M parameters in the generator and 1.5M parameters in the discriminator. In LSUN bedroom and CelebA-HQ 128×128 image generation, ResNet-19 [21] is adopted with five residual blocks in the generator and six residual blocks in the discriminator, containing about 14M parameters for each of them. Since the training of GAN is a zero-sum game between two neural networks, we remain all discriminators as the same (C3) to mitigate their influences, and replace each C3 in generators with

Table 4: Scores for different kernels. Higher inception score and lower FID is better.

| Model | C10 (IS) | C10 (FID) | LSUN (FID) | CelebA (FID) |
|---|---|---|---|---|
| C5 | 7.64±0.06 | 26.54±1.38 | 30.84±1.13 | 33.90±3.77 |
| C3 | 7.79±0.06 | 24.12±0.47 | 36.04±7.10 | 43.39±5.78 |
| C4 | 7.76±0.11 | 26.45±1.50 | 39.17±5.52 | 37.93±7.54 |
| C4sp | 7.74±0.08 | 24.86±0.41 | **24.61±2.45** | **30.22±3.02** |
| C2 | | non-convergence | | |
| C2sp | 7.77±0.05 | **23.35±0.26** | 27.73±6.76 | 31.25±4.86 |

a C4, C2, C4sp, or C2sp. Besides, the number of channels is reduced to 0.75× in C4 and C4sp, or expanded 1.5× in C2 and C2sp to approximate the same number of parameters.

The inception scores [34] and FIDs [12] are shown in Table 4 for quantitatively evaluating generated images, and examples from the best FID runs are visualized in Figure 4. Symmetric padding is crucial for the convergence of C2 generators, and remarkably improves the quality of C4 generators. In addition, the standard derivations (±) confirm that symmetric padding stabilizes the training of GANs. On CIFAR10, C2sp performs the best scores while in LSUN bedroom and CelebA-HQ generation, C4sp is slightly better than C2sp. The diverse results can be explained by the information erosion hypothesis: In CIFAR10 generation, the network depth is relatively deep in terms of image size $32 \times 32$, then a smaller kernel will have less attenuation rate and more channels. Whereas the network depth is relatively shallow in terms of image size $128 \times 128$, and the edge effect is negligible. Then larger RFs are more important than wider channels in high-resolution image generation. In summary, the symmetric padding eliminates the shifting problem and simultaneously expands the RF. The former is universal, while the latter is limited by the edge effect on some occasions.

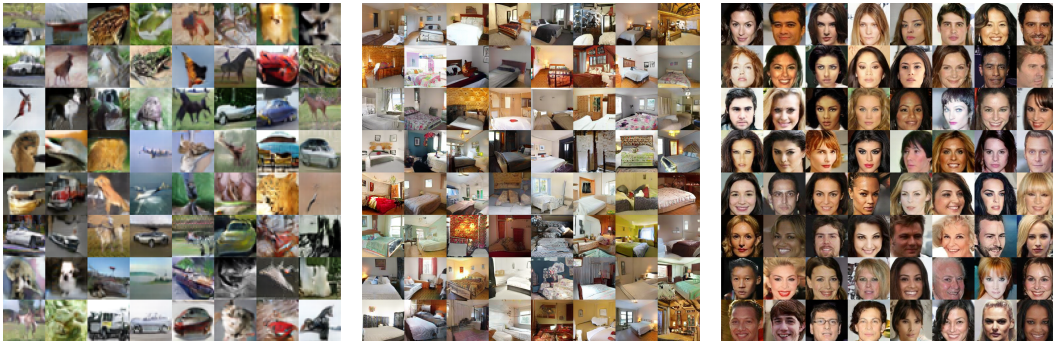

Figure 4: Examples generated by GANs on CIFAR10 ($32 \times 32$, C2sp, IS=8.27, FID=19.49), LSUN-bedroom ($128 \times 128$, C4sp, FID=16.63) and CelebA-HQ ($128 \times 128$, C4sp, FID=19.83).

## 4.5 Implementation details

Results reported as mean±std in tables or error bars in figures are trained for 5 times with different random seeds. The default settings for CIFAR classifications are as follows: We train models for 300 epochs with mini-batch size 64 except for the results in Table 2, which run 600 epochs as in [48]. We use a cosine learning rate decay [24] starting from 0.1 except for DenseNet tests, where the piece-wise constant decay performs better. The weight decay factor is 1e-4 except for parameters in depthwise convolutions. The standard augmentation [22] is applied and the $\alpha$ equals 1 in mixup augmentation.

For ImageNet classifications, all the models are trained for 100 epochs with mini-batch size 256. The learning rate is set to 0.1 initially and annealed according to the cosine decay schedule. We follow the data augmentation in [36]. Weight decay is 1e-4 in ResNet-50 and DenseNet-121 models, and decreases to 4e-5 in the other compact models. Some results are worse than reported in the original papers. It is likely due to the inconsistency of mini-batch size, learning rate decay, or total

training epochs, e.g., about 420 epochs in [35]. Our example code and models are available at `https://github.com/boluoweifenda/CNN`.

In generation tasks with GANs, we follow models and hypermeters recommended in [21]. The learning rate is 0.2, $\beta_1$ is 0.5 and $\beta_2$ is 0.999 for Adam optimizer [19]. The mini-batch size is 64, the ratio of discriminator to generator updates is 5:1 ($n_{\text{critic}} = 5$). The results in Table 4 and Figure 4 are trained for 200k and 500k discriminator update steps, respectively. We use the non-saturation loss [8] without gradient norm penalty. The spectral normalization [27] is applied in discriminators, no normalization is applied in generators.

# 5 Discussion

**Ablation study** We have tested other methods dealing with shift problem, and divided them into two categories: (1) Replacing asymmetric padding with additional non-convolution layer, e.g., interpolation, pooling; (2) Achieving symmetry with multiple convolution layers, e.g., padding 1 pixel at each side before/within two non-padding convolutions. Their implementation is restricted to certain architectures and the accuracy is no better than symmetric padding. Our main consideration is to propose a basic but elegant building element that achieves symmetry within a single layer, thus most of the existing compact models can be neatly transferred to even-sized kernels. We have also tested C3 with asymmetric padding, and observed accuracy degradation as the asymmetry gains.

**Network fragmentation** From the evaluations above, C2sp achieves comparable accuracy with less training memory and time. Although fragmented operators distributed in many groups [25] have fewer parameters and FLOPs, the *operational intensity* [39] decreases as the group number increases. This negatively impacts the efficiency of computation, energy, and bandwidth in hardware that has strong parallel computing capabilities. In the situation where memory access dominates the computation, the reduction of FLOPs does not guarantee faster execution speed [16]. We conclude that it is still controversial to (1) increase network fragmentation by grouping strategies and complex topologies; (2) decompose spatial and channel correlations by DWConvs, shift operations, and C1s.

**Naive implementation** Meanwhile, most deep learning frameworks and hardware are mainly optimized for C3, which restrains the efficiency of C4sp and C2sp to a large extent. For example, in our high-level python implementation in TensorFlow for models with C2sp, C2, and C3, despite that the parameters and FLOPs ratio is 4:4:9, the speed (SPS) and memory consumption ratio during training is about 1:1.14:1.2 and 1:0.7:0.7, respectively. The speed and memory overheads can be further optimized in the following computation libraries and software engineering once even-sized kernels are adopted by the deep learning community.

# 6 Conclusion

In this work, we explore the generalization capabilities of even-sized kernels ($2\times2$, $4\times4$) and quantify the shift problem by an information erosion hypothesis. Then we introduce symmetric padding to elegantly achieve symmetry within a single convolution layer. In classifications, C2sp achieves 30%-50% saving of parameters and FLOPs compared to C3 on CIFAR10/100, and improves accuracy for 2.5% from C2 on ImageNet. Compared to existing compact convolution blocks, C2sp achieves competitive results with fewer channels and simpler architectures, which reduce memory consumption ($>35\%$) and speed up ($>20\%$) the training process. In generation tasks, C2sp and C4sp both achieve improved image qualities and stabilized convergence. Even-sized kernels with symmetric padding provide promising building units for architecture designs that emphasize training efforts on online and continual learning occasions.

**Acknowledgments**

We thank the reviewers for their valuable suggestions and insightful comments. This work is partially supported by the Project of NSFC No. 61836004, the Brain-Science Special Program of Beijing under Grant Z181100001518006, the Suzhou-Tsinghua innovation leading program 2016SZ0102 and the National Key R&D Program of China 2018YFE0200200.

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
