[Reviews · NeurIPS 2019]

Reviewer 1



- Even-sized kernels are rarely used in models for discriminative vision tasks. Based on the results the method can effectively address this problem on image classification tasks including CIFAR-10, CIFAR-100, and ImageNet. - Additionally, some results on image generation are demonstrated. Their method consistently leads to a better FID score. - However, it's not evaluated on discriminative tasks like object detection or semantic segmentation where the spatial information is important. - Overall, the method is interesting and effective but simple.

Reviewer 2



Although this is an interesting work, the main argument of this work is not so convincing. The authors argue that the even-sized kernel will shift the feature map which results in performance degradation. However, the shifting may not be the key reason here. One popular explanation is, when using 2x2 kernel in downsampling layer with stride=2, information will be lost since no overlapping between adjacent convolution patches. The feature map shifting may not be a key issue here as convolutional operator is invariant to spatial shifts. I am not convinced that asymmetric or symmetric padding will make big difference in DCNN. ====== Post Author Feedback ====== I read the author feedback. The authors provide more experimental results to show the performance improvement of symmetric padding. I raise my score based on the new numerical results which seem promising.

Reviewer 3



Originality: Identifying and bringing up the problem of even-sized kernels is a very interesting direction to take. The method proposed in the paper is not very novel: it is in fact quite similar to what's done in [36]. However, I can also see that the authors come to the shift operation from a different direction, with the intention to address the asymmetric padding issue. I'm not aware of any work that address the asymmetric padding issue this way and I think this solution is quite novel. Quality: The experiments do not fully convince me of the claims made in the paper. More details in "Improvements". Clarity: the paper is written clearly on the methods and experiments. However the explanations and hypothesis of the "edge effect" and the "erosion" are described quite vaguely. Significance: If the experiments and results are more solid (if my concerns about experiments can be resolved), I think this would be an insightful finding in CNN architecture design. Given that the paper has novel motivation and ideas, I'm recommending a score of 6 in the hope that authors can convince me of their claims.

[Author Response · NeurIPS 2019]

**Revision summary:** We thank all the reviewers for their insightful comments. We have added experiments on ResNet/DenseNet backbones, and optimized a more efficient C2sp model for ImageNet:

Figure R1: Parameter-accuracy curves of ResNets and DenseNets.

| Model | Error (%) | Params (M) | FLOPs (M) |
|---|---|---|---|
| ResNet-50 0.5× C3 | **27.9** | 6.9 | 1127 |
| ResNet-50 0.5× C2 | 31.0 | 5.3 | 870 |
| ResNet-50 0.5× C2sp | 28.3 | **5.3** | **870** |
| ResNet C2sp optim | 26.9 | **5.8** | **573** |
| MobileNet v2 1.4× | **25.8** | 6.1 | 582 |
| ShuffleNet v2 2.0× | 27.5 | 7.4 | 591 |

Table R1: Top-1 error rates on ImageNet using the same training hyperparameters.

**Reviewer #1 Comment 1:** I would like to increase the score if the authors can show some discriminative results.

**Response:** We have tried our best to implement and train C2sp + Faster-RCNN models on MSCOCO, it's still on-going and hard to complete in such limited time. Nevertheless, we'd like to validate it through two evidences: (1) GAN training is also very sensitive to spatial information and transformations, and our results (main text Table 3) have significantly exposed the asymmetric problem. When trained with C3 discriminators, C2 generators directly lead to non-convergence, and C4sp generators are much better than C4. (2) A paper [1] replaces Conv3D with *temporal shift* + Conv2D, which achieves efficient video understanding and also generalizes to other modalities, e.g., optical flow.

**Reviewer #2 Comment 1:** The shifting may not be the key reason here. When using 2x2 kernel in downsampling layer with stride=2, information will be lost since no overlapping between adjacent convolution patches.

**Response:** This should not be a problem. In CIFAR100, the downsampling stage of DenseNet is performed by Avg-pool with $2\times2$ window (non-overlapped pooling is popular) and stride=2, not by Conv stride=2. In ResNet-50 ImageNet, all Convs stride=2 are replaced with Avg-pools + Convs stride=1, as suggested in [2]. As shown in Figure R1 and Table R1, C2sp still outperforms C2 since shifting aggravates the *information erosion* at edges (main text Line 121). Additionally, we further address the reviewer's concern by replacing C2/C2sp with C3 when stride=2 (overlapped patches). The error rates (%) are 7.33±0.11 (C2) and 6.62±0.15 (C2sp) on ResNet-38 CIFAR10, which are consistent with Figure R1.

**Reviewer #2 Comment 2:** CIFAR10/100 use different backbones. Report: (1) the performances of ALL backbones with C2, C2sp, C4 and C4sp. (2) the performance improvement of symmetric padding against network depth.

**Response:** The performances of ALL backbones are shown in Figure R1. The accuracy gaps between C2 and C2sp are larger in deeper networks. As for C4 and C5, we have claimed (main text Line 172) that the degradation is dominated by "edge effect", so C4sp only slightly improves the accuracy (but significantly in GANs). Different backbones can cross-validate the generality and consistency since architectures may affect the results (e.g., concerns in Comment 1).

**Reviewer #3 Comment 1:** To address this concern of cherry-picking, I recommend the authors to explain which channels are selected and show more channels in a figure.

**Response:** They are not cherry-picked. Since each single channel is very stochastic and hard to interpret (examples in Figure R2, 9 channels for 3 stages, C2), the activations in Figure 1 are the average values of all channels, i.e., 16, 32, and 64 channels.

**Reviewer #3 Comment 2:** An experiment using C3 with asymmetric padding.

**Response:** We test ResNet-38 (#channel 18-36-72) on CIFAR10 with four settings: C3 {1111}, C3sp (9 symmetric groups {0202}, {0211},..., {2020}), C3ap3 (3 asymmetric groups {0211}, {1102}, {0202}) and C3ap1 {0202}. The error rates (%) are 5.51±0.08, 5.94±0.05, 6.21±0.03 and 7.27±0.28. The asymmetry gains, the accuracy degrades. C3sp has expended RF 5×5 and is restricted by the "edge effect", as with C4&C5.

Figure R2: colormaps

**Reviewer #3 Comment 2:** The performance degradation in C4 is dominated by "edge effect" rather than the shift, I recommend the authors to provide more convincing arguments on their issues, e.g., perhaps by comparing with C5.

**Response:** In Figure R1, error rates C5≫C4>C4sp≫C3, which is consistent with the "edge effect". Although C4sp provides minor improvement in classifications, it is much better than C4 in GANs, where the "edge effect" is negligible regarding the network depth and image resolution. In summary, the symmetric padding eliminates the shifting problem, and simultaneously expands the reception field. The former is critical, the latter is limited on some occasions.

[1] Lin, Ji, et al. "Temporal shift module for efficient video understanding." *arXiv:1811.08383* (2018).
[2] Zhang, Richard. "Making convolutional networks shift-invariant again." In *ICML*. 2019.


[Meta-Review · NeurIPS 2019]

This paper proposes a technique to improve convolutional neural networks. The technique relies on the use of symmetric padding to address the shift problem in even-sized convolutions. The reviewers found the method to be sound and the experimental validation on CIFAR-10, CIFAR-100, and ImageNet to be convincing. The concerns raised by the reviewers were later addressed by the rebuttal.